# Assessing and Mapping Changes in Forest Growing Stock Volume over Time in Bashkiriya Nature Reserve, Russia

**Larisa Belan** [1], **Azamat Suleymanov** [1], **Ekaterina Bogdan** [1,*], **Aleksandr Volkov** [2], **Ildar Gaysin** [2,3], **Iren Tuktarova** [1] **and Ruslan Shagaliev** [2]

[1] Department of Environmental Protection and Prudent Exploitation of Natural Resources, Ufa State Petroleum Technological University, Kosmonavtov Str., 1, 450064 Ufa, Russia; belan77767@mail.ru (L.B.); filpip@yandex.ru (A.S.); umrko@mail.ru (I.T.)

[2] Decarbonisation Technologies Center, Ufa State Petroleum Technological University, Kosmonavtov Str., 1, 450064 Ufa, Russia; volkovufa@mail.ru (A.V.); i.gaisin2012@yandex.ru (I.G.); shagaliev@rambler.ru (R.S.)

[3] Federal State Budgetary Institution "Bashkiriya State Nature Reserve", Karat Str., 2, 453580 Starosubkhangulovo, Russia

* Correspondence: eavolkova@bk.ru

**Abstract:** There is growing recognition that forest ecosystems are a key component in the global carbon cycle, and there is a clear demand for their study. This research is a comparative analysis of forest growing stock volume (GSV) and determination of annual growth in Bashkiriya Nature Reserve (Russia) for 1979 and 2015 using 8395 and 8405 observation plots, respectively. Also, we evaluated the spatial distribution and produced digital maps of the species and their GSV for each year. The results showed that pine and birch were the dominant species (60.5 and 24.8% of the area in 2015, respectively) and there were no significant changes in the area of stands during the 36-year period. We found that the GSV in the reserve had increased by an average of 23.2% over the 36-year period. Specifically, the total forest GSV increased from 7,678,960 in 1979 to 10,003,890 $m^3$ in 2015, representing an annual gain of 0–1.5 $m^3$/ha. The increase in GSV was mainly associated with an increase in birch and pine trees. The annual growth of GSV was determined as 1.8–2.1 and 1.4 $m^3$/ha per year for pine and birch forests, respectively. However, these types of trees belong to the age categories of mature and overmature stands, i.e., with reduced intensity of GSV gain. Digital maps produced as part of this study provide a visual representation of the changes in forest spatial patterns and GSV over time, highlighting areas of the reserve where the stock has increased or decreased. This study leveraged a substantial dataset, which provided valuable retrospective insights into the dynamics of pristine forest ecosystems, allowing for the assessment of changes over a 36-year period. Overall, this study highlights the importance of the ongoing monitoring and assessment of GSV levels, especially in the context of rapidly changing environments and climates.

**Keywords:** forest growing stock volume; mapping; space–time; climate change

## 1. Introduction

Forests are crucial for absorbing and storing carbon dioxide from the atmosphere and therefore are a key component of the biosphere to mitigate the effects of climate change [1,2]. Forest growing stock volume (GSV) is an important indicator of a forest's ability to sequester carbon and its assessment is an important task worldwide. At the same time, accurately measuring GSV is essential for sustainable forest management as it informs decisions about harvesting, conservation, and reforestation efforts [3]. Moreover, forests are home to a wide variety of flora and fauna and a healthy amount of GSV ensures that a forest can provide suitable habitats and resources for various plant and animal species [4].

Comparing forest GSV values for different periods is important because it provides insights into changes in forest structure and composition over time [5–7]. In particular, such studies can help us understand the role of forests in mitigating climate change. Finally,

monitoring changes in planted forests provides an opportunity to better understand the extent to which forests absorb carbon and identify strategies to increase their carbon sequestration potential.

Digital maps of the qualitative and quantitative characteristics of forests should be used to monitor and manage forest resources. Also, this includes tracking forest cover changes and identifying areas of high biodiversity and carbon sequestration potential. With regard to climate change, monitoring and mapping GSV over different periods is becoming an essential task. Currently, various methods are used to determine GSV [3], and they can be mainly categorized as (1) sample tree measurements (including tree species, diameter at breast height, upper diameter, tree height, and more), (2) volume models (estimating the sample tree volumes based on the previously described field measurements), and remote sensing technologies. Despite the fact that remote sensing data and machine learning methods are successfully used in GSV spatial assessment today [8–10], the correct implementation of these methods is severely limited without ground-based data. Thus, conventional field methods are necessary to establish relationships with explanatory variables (e.g., remote sensing data). This is especially important for previously unexplored areas since there are no data for training statistical and mathematical models. In general, field measurements are highly detailed and accurate, making them particularly useful for the training and validation of machine learning models. Nevertheless, it should be noted that conventional methods have limitations, including resource and time requirements.

Currently, global forests are experiencing added stress due to climate change, which has a significant impact on GSV levels. Gschwantner et al. [3] emphasized in a review article that, to inform the future development of a knowledge-based bioeconomy, there is a demand for the maintenance, extension, and harmonization of existing forest databases. This is essential to enable the analysis of forest ecosystem changes on a large spatial and long temporal scale. In this regard, retrospective monitoring is a crucial solution for tracing changes in forest characteristics in both space and time. Thus, the main objectives of this study were (1) to determine the dominant species in the Bashkiriya Nature Reserve (Russia) for 1979 and 2015 and their changes, (2) to estimate GSV values for each of the years and annual growth, and (3) to generate digital maps for each year under study. Producing digital maps of GSV for these two periods is also important because it allows us to visualize and analyze the changes in forest structure and composition over time. By creating these maps, we can identify areas where forest growing stock has increased or decreased and assess the potential implications of these changes for biodiversity and carbon sequestration.

## 2. Materials and Methods

### 2.1. Study Area

This study was conducted in the Bashkir State Nature Reserve—a specially protected natural area of federal importance. The reserve is located in the central part of the Southern Urals mountains and occupies an area of 496 km$^2$ (Figure 1). The territory is characterized by the transition from a mountain–forest to a steppe–forest zone. The climate is sharply continental. The average annual temperature is +6 °C, the absolute maximum temperature is +35.9 °C, and the minimum temperature is −48.6 °C. The average annual precipitation is 587 mm/year. South-westerly and westerly winds prevail.

The reserve encompasses two ridges: Southern Kraka and Uraltau, the vegetation of which is different. Southern Kraka is located in the western part of the reserve, adjacent to the southern portion of mid-altitude mountain ranges ranging in elevation from 750 to 1034 m. It represents a partially isolated mountain node to the west of the main central uplifts of the Southern Urals. The area is dominated by mixed light forests with a prevalence of Scots pine (*Pinus sylvestris*) and Siberian larch (*Larix sibirica*) [11], which, on the slopes of southern exposures, along with the wide distribution of mountain steppes, form rare communities [12]. The Uraltau ridge, located in the eastern half of the reserve's territory, is separated from the Southern Krakka by the South Uzian River. Here, birch (*Betula pendula*) and aspen (*Populus tremula*) species prevail [13]. Forests of the reserve belong to the zone

of boreal pine–birch forests and cover more than 80% of the reserve area. The mountain forests of the reserve are specially protected.

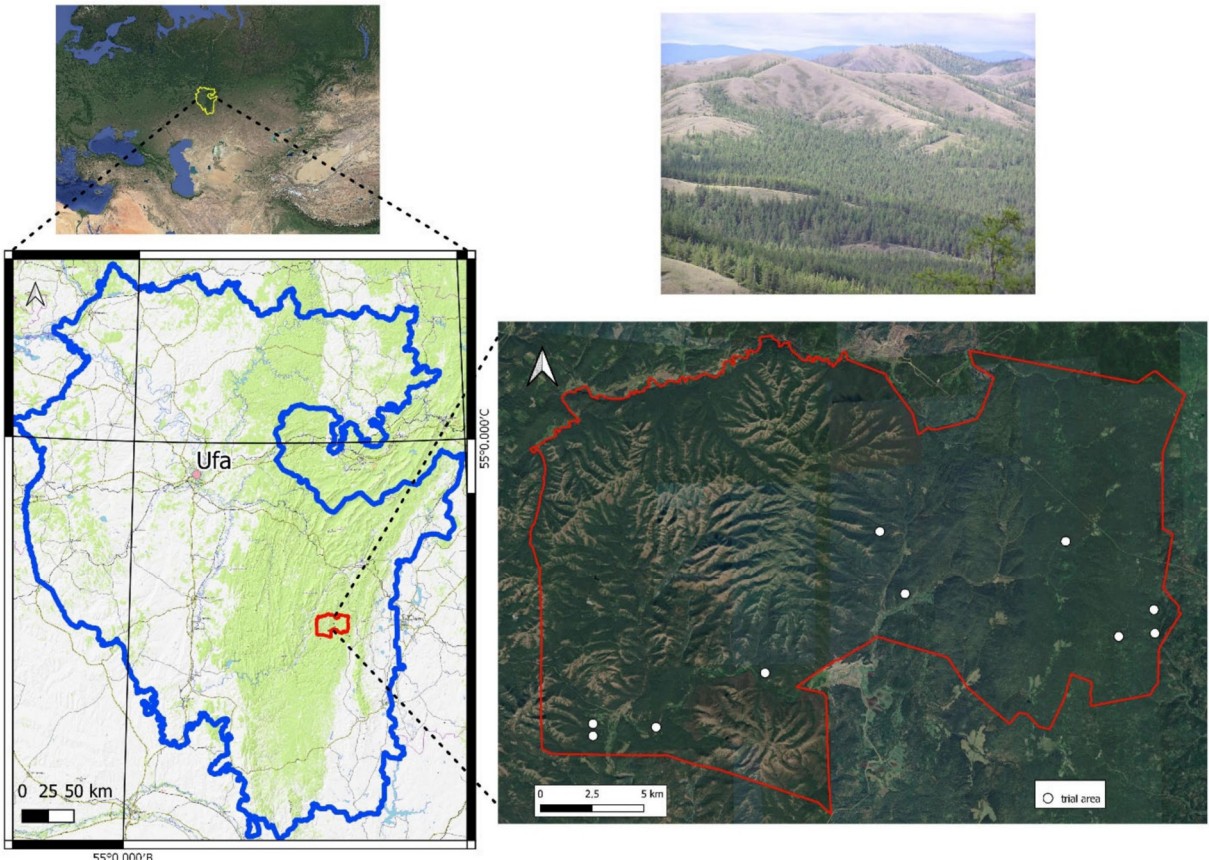

**Figure 1.** Location of the Republic of Bashkortostan and Bashkiriya State Nature Reserve.

The designation of the territory as a "state natural reserve" implies a complete cessation of economic use. Only scientific observations of ecosystem changes are permitted. However, it should be noted that the reserve was closed in 1951 and resumed operation only in 1958. During this period, a forestry enterprise operated within its boundaries, with timber harvesting as one of its responsibilities. Additionally, cattle grazing took place on the mountainous steppes, parts of the territory were plowed for agriculture, and extensive areas were used for haymaking. Nevertheless, in 1958, the reserve status was reinstated, and, today, only scientific observations are conducted within its territory.

### 2.2. Field Investigation

This study used the materials of forest taxation in 1979 and 2015. Furthermore, to update the data, we conducted forest inventory work on 10 sample plots in the summer of 2023 (Figure 1). Thus, the number of survey sites was 8395 in 1979 and 8405 in 2015 (Figure 2). The taxation characteristics were performed according to the manual of Anuchin [14]. To study the organic mass of forest phytocenosis, one trial plot was established in each forest area. At this site, measurements and observations were made for growth, decay, microclimate, etc., without disturbing the original condition of the ecosystem. An eye-measuring study of the tree characteristics was carried out. The diameter of the tree was measured using a tree caliper, which consists of a measuring ruler with a scale of two parallel bars plotted to it. To determine the height of the tree, first, distances of 10, 15, and 20 m were measured from it. Then, using one of these distances through diameters, sighting on the top of the tree was carried out. Next, the angle between the horizontal position and the line of sight was counted using an eclimeter and the tree height was determined.

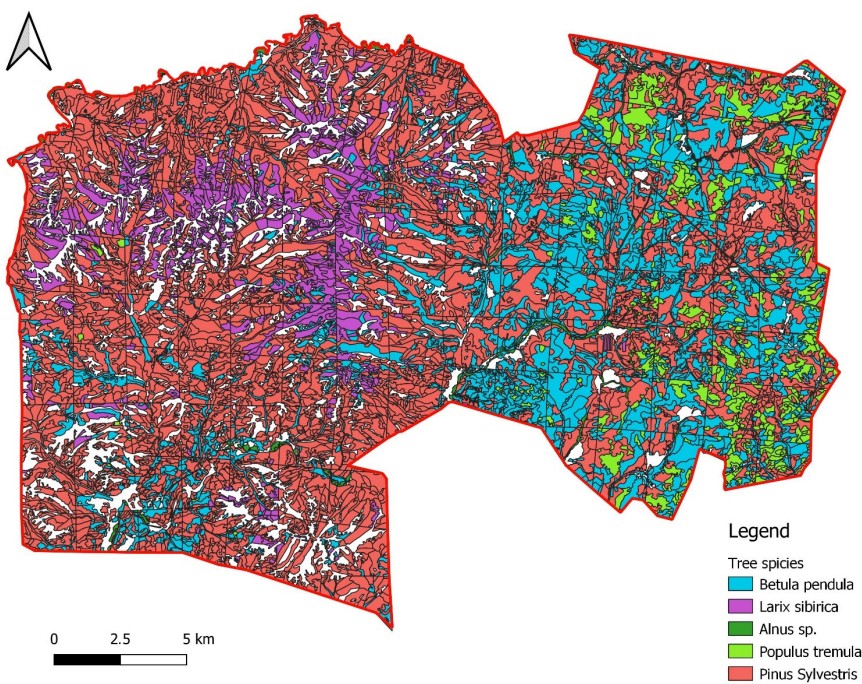

**Figure 2.** Spatial distribution of forest plots of the reserve by dominant tree species.

Forest GSV values for each plot were determined based on the average values of height, diameter, and stand density (a parameter derived from the number of trees per 1 ha) in each section, using auxiliary tables from the reference manual "All-Union standards for forest taxation" [14]. This manual includes information (growing stock by dominant species, age, site indices, stocking) on all forests in Russia and former Soviet republics. In the control areas, GSV values for each tree were calculated based on a height and diameter at a height of 1.3 m, according to Equation (1). For each control area, the GSV value was summed and then divided by the area in hectares:

$$V = H \times G \times F \tag{1}$$

where V is the volume of the tree stand ($m^3$); H is the height of the tree (m); G is the tree cross-sectional area ($m^2$), calculated using the values of tree diameter; and F is the trunk shape factor from the auxiliary tables [15].

### 2.3. Digital Mapping and Statistical Analyses

Digital mapping of dominant species and GSV values for each plot according to the studied periods, as well as the statistical analysis, was performed in QGIS 3.16.1. To do this, a vector file was prepared with the boundaries of taxation plots covering the entire area of the territory. Then, the obtained qualitative (tree species) and quantitative (GSV) parameters were recorded in the attribute table of vector files and visualized.

## 3. Results

### 3.1. Forests Characteristics and Its Changes

Figure 2 shows the spatial distribution of predominant tree species in 2015. Since the visual changes between 1979 and 2015 are not noticeable, we show the actual distribution of species for 2015. According to the visualization, pine (*Pinus sylvestris*) and larch (*Larix sibirica*) species predominated in the western and central parts, while birch (*Betula pendula*) and aspen (*Popula tremula*) were widespread in the eastern part of the reserve.

The territory of the reserve can be divided into two parts: the Southern Krakka Range (63% of the total area) and the Uraltau Range (37%). The forests in these ranges exhibit some differences. An analysis of the distribution of tree species by age groups and GSV in the

Southern Krakka Range (Table 1) revealed that more than half of the range's territory (59.8%) is covered by mature pine forests (80–120 years old). These mature pine forests contributed to the largest increase in GSV, accounting for 1322.5 million m$^3$, which constitutes 90.8% of the total GSV in the range. Spruce and birch species occupy approximately equal areas, representing 13.6% and 12.6% of the total range's territory, respectively. Notably, birch forests showed an increase in GSV by 81.0 million m$^3$, primarily due to mature stands aged 61–135 years. In contrast, mature spruce forests demonstrated a decrease in GSV by 1.9 million m$^3$ since 1979.

**Table 1.** Characteristics of forest species and their changes for the period 1979–2015 in the Southern Krakka Range.

| Age | Area, km$^2$ | GSV, m$^3$ 2015 | Difference, m$^3$ 2015–1979 | Proportion of Total Area, % | Proportion of Total GSV, % |
|---|---|---|---|---|---|
| Birch (*Betula pendula*) | | | | | |
| 20–30 | 116.9 | 13,622.24 | 9060.69 | 0.4532 | 0.0027 |
| 31–40 | 161.9 | 24,373.72 | 23,093.25 | 0.6277 | 0.0049 |
| 41–50 | 79.6 | 10,353.97 | 7028.68 | 0.3086 | 0.0021 |
| 51–60 | 61.5 | 15,212.95 | 6689.91 | 0.2384 | 0.0030 |
| 61–135 | 2841.6 | 25,415,452.26 | 8,051,196.21 | 11.0170 | 5.0932 |
| Larch (*Larix sibirica*) | | | | | |
| 20–40 | 131.1 | 27,500.45 | 25,009.55 | 0.5083 | 0.0055 |
| 60–80 | 74.3 | 22,169.19 | 10,180.81 | 0.2881 | 0.0044 |
| 81–100 | 522 | 1,108,896.61 | 347,536.64 | 2.0238 | 0.2222 |
| 101–120 | 509.02 | 815,590.02 | 755,774.06 | 1.9735 | 0.1634 |
| 121–140 | 280.3 | 1,786,113.65 | 1,643,219.23 | 1.0867 | 0.3579 |
| 141–280 | 1995.3 | 6,140,619.55 | −1,919,769.91 | 7.7358 | 1.2306 |
| Alder (*Alnus* sp.) | | | | | |
| 20–30 | 60.4 | 5241.15 | −4897.90 | 0.2342 | 0.0011 |
| 31–40 | 147.3 | 35,943.70 | −12,759.86 | 0.5711 | 0.0072 |
| 45 | 9.7 | 133.85 | 20.14 | 0.0376 | 0.0000 |
| 70 | 6.7 | 27.00 | −2.77 | 0.0260 | 0.0000 |
| Aspen (*Populus tremula*) | | | | | |
| 70–115 | 60.8 | 17,420.90 | 4140.72 | 0.2357 | 0.0035 |
| Pine (*Pinus sylvestris*) | | | | | |
| 20–40 | 202.7 | 52,714.57 | 45,830.47 | 0.7859 | 0.0106 |
| 41–60 | 148.8 | 102,030.08 | 58,714.10 | 0.5769 | 0.0204 |
| 61–80 | 1099.6 | 6,272,568.14 | 2,777,934.87 | 4.2632 | 1.2570 |
| 81–100 | 7292.4 | 207,951,835.87 | 72,291,369.72 | 28.2729 | 41.6728 |
| 101–120 | 8117.8 | 243,773,515.69 | 59,962,697.95 | 31.4730 | 48.8513 |
| 121–140 | 520.1 | 1,031,592.87 | 261,268.07 | 2.0164 | 0.2067 |
| 141–280 | 1353.1 | 4,388,212.90 | 246,061.24 | 5.2460 | 0.8794 |

In the Uraltau Range, mature pine stands aged 80–120 years make up 44.5% of the forest cover (Table 2), which is less than in the Southern Krakka Range. However, the increase in the volume of this type of forest since 1979 was significantly lower in the Uraltau Range, totaling 137.9 million m$^3$. This increase was an order of magnitude lower than that observed in the Southern Krakka Range. Birch is the second most prevalent tree species, covering 26.8% of the Uraltau Range. The volume of birch species has increased by 17.4 million m$^3$ since 1979, which is 4.6 times lower than the increase observed in the Southern Krakka Range. In contrast to the Southern Krakka, aspen covers 17.4% of the Uraltau and showed an increase in the total volume of mature stands (60–130 years) by 8.7 million m$^3$. In summary, despite a 1.7 times difference in the area of the Southern Krakka and Uraltau Ranges, the total volume of the main tree species (birch and pine)

exceeded that in the Southern Krakka by 4.6 to 10 times for mature forests. The analysis of the results for young and middle-aged pine and birch stands revealed a similar trend.

**Table 2.** Characteristics of forest species and their changes for the period 1979–2015 in the Uraltau Range.

| Age | Area, km$^2$ | GSV m$^3$ 2015 | Difference, m$^3$ 2015–1979 | Proportion of Total Area, % | Proportion of Total GSV, % |
|---|---|---|---|---|---|
| | | Birch (*Betula pendula*) | | | |
| 30–40 | 4.6 | 158.81 | 143.63 | 0.0370 | 0.0002 |
| 41–50 | 5.2 | 104.94 | 93.40 | 0.0418 | 0.0001 |
| 51–60 | 2.3 | 23.00 | 14.64 | 0.0185 | 0.0000 |
| 61–150 | 3324.9 | 8,686,612.19 | 1,739,105.22 | 26.7222 | 9.6660 |
| | | Larch (*Larix sibirica*) | | | |
| 47–50 | 1.0 | 19.00 | 18.17 | 0.0080 | 0.0000 |
| | | Alder (*Alnus* sp.) | | | |
| 0–20 | 3.2 | 16.00 | 8.00 | 0.0257 | 0.0000 |
| 21–30 | 35.3 | 883.03 | −326.98 | 0.2837 | 0.0010 |
| 31–40 | 43.2 | 1309.74 | 573.26 | 0.3472 | 0.0015 |
| 60–70 | 4.4 | 191.22 | 47.14 | 0.0350 | 0.0002 |
| | | Aspen (*Populus tremula*) | | | |
| 0–10 | 19.4 | 56.70 | −1787.93 | 0.1558 | 0.0001 |
| 60–130 | 2159.5 | 5,600,350.44 | 870,074.49 | 17.3559 | 6.2318 |
| | | Pine (*Pinus sylvestris*) | | | |
| 40–60 | 42.3 | 8074.90 | 6969.94 | 0.3400 | 0.0090 |
| 61–80 | 57.0 | 19,651.61 | 9993.87 | 0.4581 | 0.0219 |
| 81–100 | 1222.2 | 6,962,712.07 | 1,941,623.59 | 9.8228 | 7.7477 |
| 101–120 | 4321.0 | 66,917,615.88 | 11,844,707.92 | 34.7279 | 74.4621 |
| 121–140 | 531.6 | 1,014,756.89 | 135,041.28 | 4.2725 | 1.1292 |
| 141–220 | 665.4 | 655,437.47 | 77,278.06 | 5.3478 | 0.7293 |

The analysis of the dynamics of forested areas since 1979 revealed that the total forested area increased by 1.894 hectares. Simultaneously, mountain steppes expanded by 394.5 hectares, while the area occupied by other open habitats (meadows, clearings, wastelands) decreased by 2.362 hectares. The area of land transformed by fires also decreased by 1.609 hectares. From these findings, it can be concluded that the overall increase in forested areas is attributed to the natural reforestation of previously burnt forested areas, meadows, and clearings, among others, rather than to the expansion of steppes. Furthermore, it is highly likely that the expansion of steppes is a result of the reduction in the area of forested regions dominated by deciduous trees.

Figure 3 shows the spatial distribution of GSV values for both years. The highest frequency was found in areas with minimal GSV values, whereas the largest distribution was found in the range from 190 to 350 m$^3$/ha. Despite the heterogeneity of forest stands, which is reflected in the multimodality of the histogram of the GSV distribution, the interval of the highest frequencies of occurrence of GSV was within the values of 200–300 m$^3$/ha.

Since pines were the dominant species, they were characterized by the largest GSV–5,035,160.0 m$^3$ in 1979 and 6,783,110.0 m$^3$ in 2015, which has increased by 1,747,950 m$^3$ over a 36-year period. During the same period, birch forests gained 478,550 m$^3$ of GSV, while larch, aspen, and alder showed gains of 45,060, 45,630, and 7740 m$^3$, respectively. We also found that during this period, an increase in GSV of 2,324,930 m$^3$ was found, which amounted to a total of 10,003,890 m$^3$.

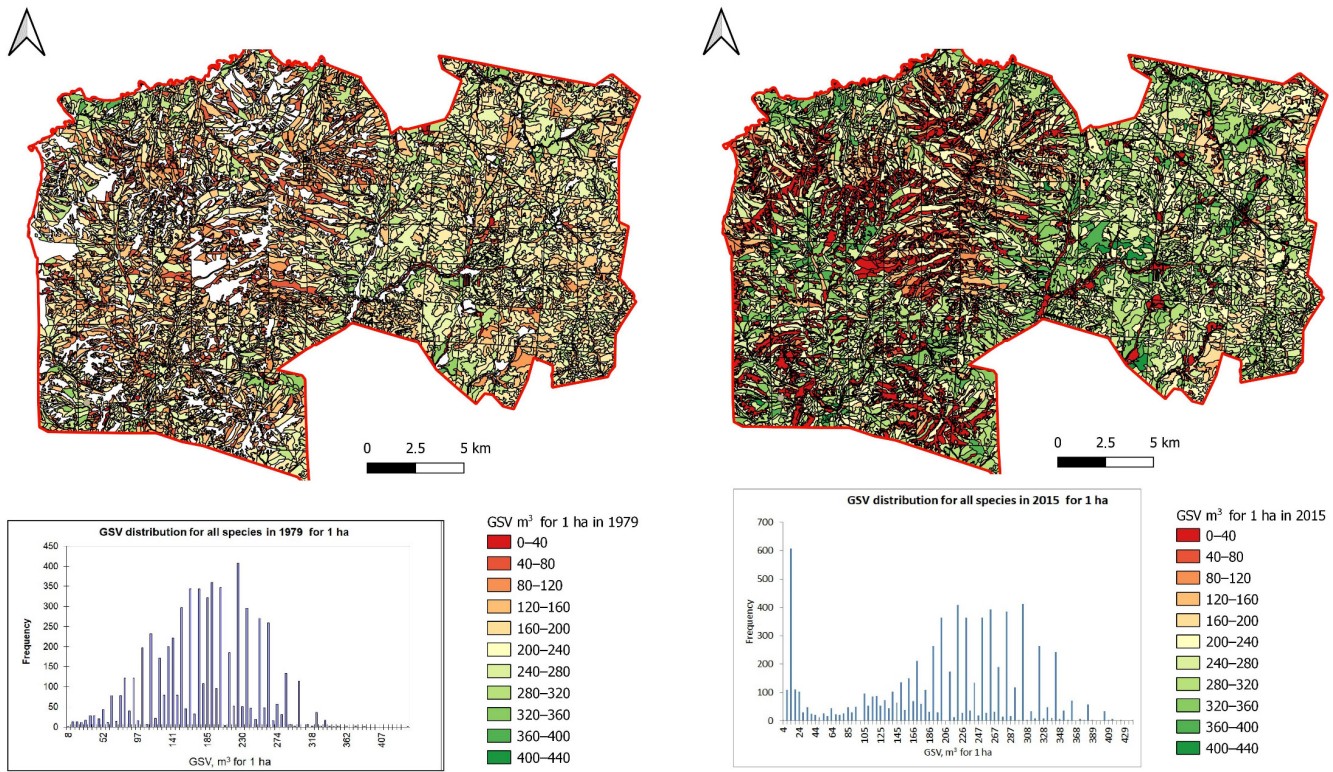

**Figure 3.** Spatial distribution of GSV by study plots in 1979 (**left**) and 2015 (**right**).

### 3.2. Changes in GSV Rates

The resulting GSV gain map (Figure 4) was produced by subtracting two maps for the studied years. The map shows that in a large part of the reserve, the annual growth values were close to zero or negative. The maximum values of growth were recorded at sites 76, 96, and 119, located in the western part of the territory, within the Southern Kraka mountain range. In this area, forest growing conditions are worse than in the eastern part (Uraltau ridge). Figure 3 also shows the histogram of the annual increase in GSV values by reserve plots for the 1979–2015 period. This histogram was calculated by the difference in GSV in each study plot and divided by 36 (the number of years between forest studies). The distribution of annual growth values shows notable heterogeneity and the presence of several modal values, but the main mode of growth lay in the range of 0.0 to 1.5 m³/ha per year.

The analysis allows us to draw preliminary conclusions about the presence of the main classes of forests in terms of carbon deposition potential. These classes were conventionally indigenous pine, birch–pine, and birch forests. Pine forests are characterized by the potential of carbon absorption, expressed in the values of the annual growth of GSV of 1.8–2.1 m³/ha per year (Figure 5), whereas these values are 1.4 m³/ha per year for birch forests (Figure 6).

The analysis of the inventory results of sample plots showed that the growth rate for pine stands from 2015 to 2023 was 0.95 m³/ha per year. In contrast, the average annual growth rate for these sample plots from 1979 to 2015 was 1.35 m³/ha. This indicates a decrease in the productivity of pine stands in the reserve.

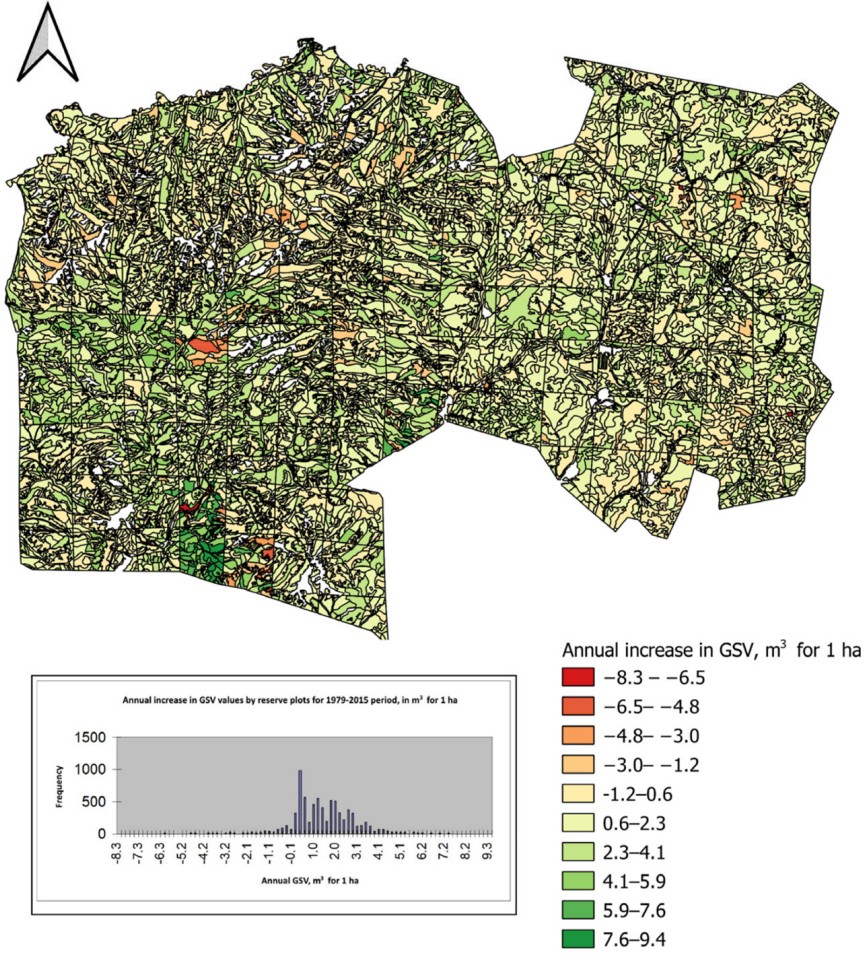

**Figure 4.** Spatial distribution of annual GSV growth by plots for the period 1979–2015.

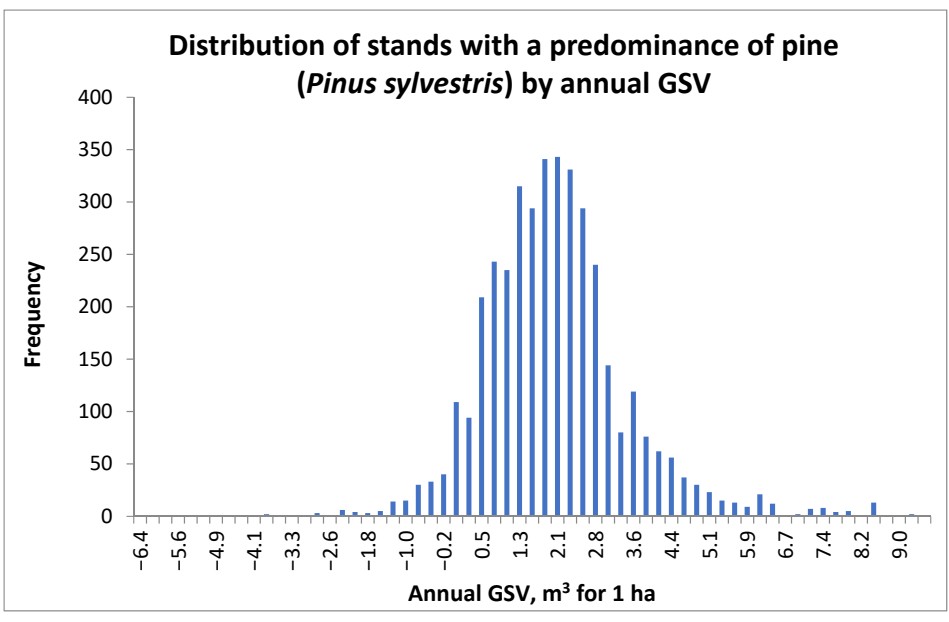

**Figure 5.** Distribution of annual growth of stands with a predominance of pine.

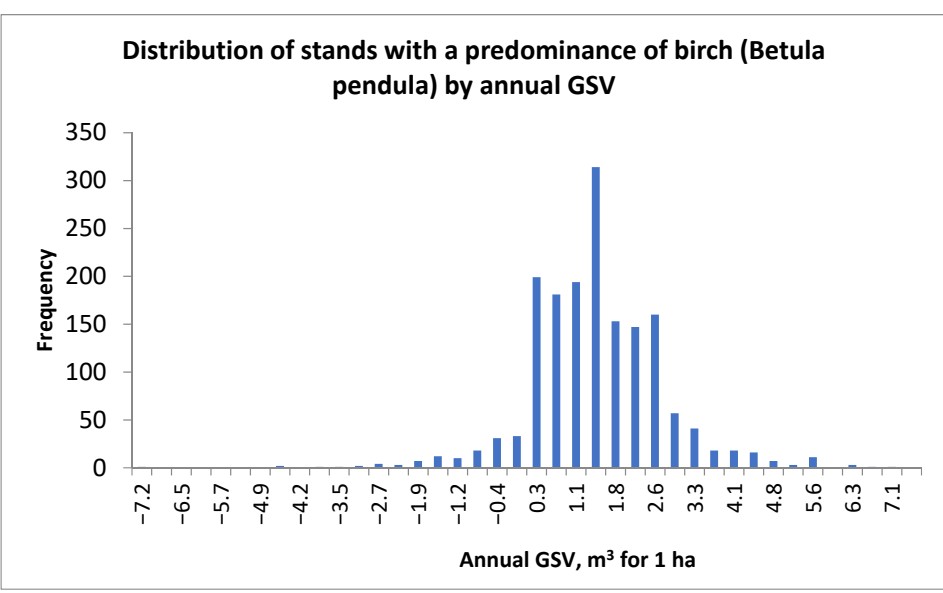

**Figure 6.** Distribution of annual growth of stands with a predominance of birch.

### 3.3. Age Structure of Forests

Figures 7–9 show histograms of the distribution of all stands of the reserve by age. According to the results, the age of the trees mainly varied from 80 to 125 years. The most common ages of pine stands were 85–120 years old, while birch stands were 70–120 years old. Thus, these types of trees belong to the age categories of mature and overmature stands, i.e., with reduced intensity of growth.

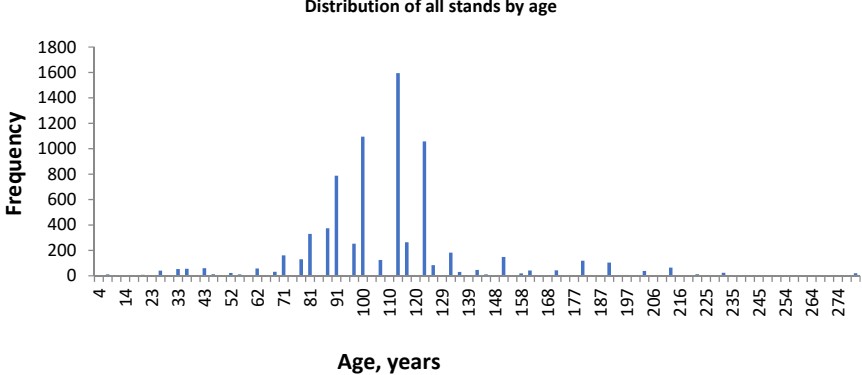

**Figure 7.** Histogram of the distribution of stands by age.

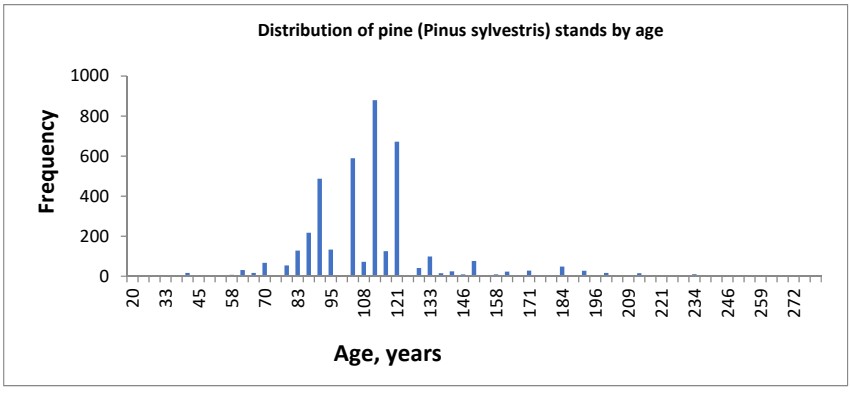

**Figure 8.** Histogram of the distribution of pine stands by age.

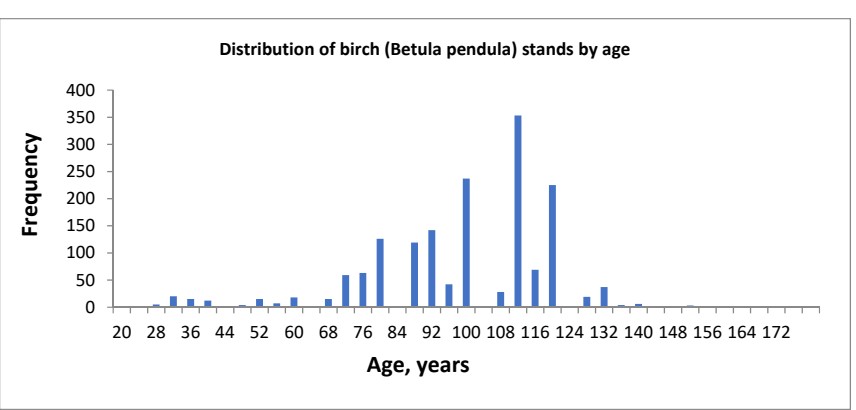

**Figure 9.** Histogram of the distribution of birch stands by age.

## 4. Discussion

### 4.1. Forests of the Reserve

Forests in well-managed reserves and protected areas are typically characterized by a high degree of biodiversity and a lower level of human disturbance compared to other areas [16]. Trees in such territories may have fewer stressors and competition, which can contribute to faster growth rates and overall healthier trees than forests in unprotected or disturbed areas. For example, Zamolodchikov et al., 2011 [17], reported that the carbon sink of Russian forests has increased from 80 Mt C $\times$ yr$^{-1}$ in 1988 to 230–240 Mt C $\times$ yr$^{-1}$ in the late 2000s due to the decline in harvesting and fire management, which started in the 1990s.

The Bashkiriya Nature Reserve was founded in 1929 and, from that moment, the anthropogenic influence was either completely excluded or minimized. However, there are also factors that can disrupt tree growth in these areas. Natural disturbances such as storms, fires, and disease outbreaks are relatively common in the region and in the reserve, which also affect tree growth and mortality [18–20]. The above-mentioned negative factors are also exacerbated by climate change, which has been established in the study region [21–23]. Changes in temperature, precipitation patterns, and other climatic factors can impact the availability of resources and create new challenges for forest managers [24].

Earlier, studies were conducted to study the dynamics of woody vegetation in the territory of the Southern Krakka ridge in the Bashkir State Nature Reserve [25,26]. It has been established that, as a result of the increase in temperature and precipitation in the Southern Urals in recent decades, which was most observed in winter, there was an advance of the forest boundary and an increase in the closeness of stands in the forest–mountain steppe ecotone. These changes on the modern border of closed forests led to an increase in aboveground phytomass by 32.8–56.6 t/ha. Our studies confirm this conclusion and also demonstrate an increase in phytomass in the Uraltau ridge.

### 4.2. Carbon Sequestration by Tree Species

The largest reserves of forest carbon (50–60 t/ha) and the maximum annual carbon deposition (3.5–4.2 t/ha) are located in the mountainous and foothill regions of the Republic of Bashkortostan [27]. In the reserve, pine and birch forests made the main contribution to the deposition of carbon in the reserve territory. Moreover, among these dominant tree species, the contribution of pine trees to carbon sequestration was about three times greater than that of birch trees. This conclusion follows from a comparison of the total increase in the GSV values for the period 1979–2015 since the increase in the GSV values amounted to 478,550 m$^3$ in birch forests and 1,747,950 m$^3$ in pine forests. The increase in GSV values in plots with a predominance of larch and aspen was almost 38 times less compared with the increase in pine forests. The main reason for such a low annual growth of forests is due to the predominance in the reserve of forests belonging to the mature and overmature age group, i.e., old-aged forests characterized by a natural decrease in the intensity of growth.

Moreover, similar results were found in other parts of the Republic of Bashkortostan. Volkov et al. [28] conducted an analysis of changes in GSV in the territory of Kandry-Kul Park and established a greater potential for depositing and increasing phytomass in pine stands than in birch stands. In general, the dominant tree species in the reserve are pine and birch. In the reserve, pines are conventionally indigenous plantings that arose naturally, probably after logging and fires, but that are self-seeding on the site of spruce forests that existed there in previous centuries. Other tree species (larch, aspen, gray alder) in the reserve have much smaller areas and proportions in the total area of forests than pine and birch. According to Tables 1 and 2, there was a slight decrease in the area occupied by plots, with a predominance of larch and aspen.

### 4.3. Further Prospects for the Digital Mapping of Tree Species and GSV

In the future, addressing the limitations of conventional methods for GSV assessment could involve embracing technological advancements and innovative approaches. For instance, our methods of digital mapping tree species and GSV values were based on integration with geographic information systems (GIS). However, remote sensing data and unmanned aerial vehicles (UAV) have made significant progress in the digital mapping of tree species and GSV [29]. Thus, the use of remote sensing data has shown promising results in the assessment and mapping of forests and their characteristics [30,31]. Moreover, light detection and ranging (LiDAR) technology provides higher accuracy of spatial prediction than traditional remote sensing data due to its capability of gathering a three-dimensional point cloud of a forest environment in a short time [32]. For example, Brilli et al. [33] successfully used LiDAR data for GSV assessment in the main urban park of Florence. Raciti et al. [34] demonstrated the application of LiDAR to map canopy cover and aboveground tree carbon storage in Boston, USA.

The correlation between field-based observations and remote sensing data is established using a variety of methods, ranging from simple regressions to machine learning methods [35]. Machine learning algorithms have been used to classify different types of vegetation based on spectral data from remote sensing and UAV sensors. The accuracy of these algorithms can be improved by incorporating more training data, optimizing feature extraction, and using more sophisticated algorithms such as deep learning [36].

Also, one key solution is to apply volume models, which estimate sample tree volumes based on previously described field measurements. This approach can significantly reduce the time and resources required for extensive fieldwork in similar ecosystems. Thus, our detailed field surveys can form the basis for the development of spatial predictive models using remote sensing data and machine learning methods. Such investigations allow us to extrapolate the results received in this work on the territory with similar tree species and environmental conditions. Furthermore, having a dataset spanning different years (1979 and 2015) and collected using the same methodology, it is possible to utilize it for modeling GSV in both space and time, enabling us to forecast future changes. These approaches, combined with ongoing research and technological advancements, offer promising avenues for overcoming the limitations of traditional methods in forest GSV assessment.

### 4.4. The Prospect of Applying GSV Results

The results of the GSV assessment in the Bashkiriya Nature Reserve hold application prospects that are integral to the sustainable management and conservation of this valuable natural resource. First, the GSV results can guide sustainable forest management practices within the reserve. This information can inform decisions about selective logging, afforestation, and reforestation, ensuring that forest resources are used in an environmentally responsible manner while maintaining long-term forest health. Moreover, the reserve is known for its rich biodiversity, with numerous endangered species and unique ecosystems [37]. For instance, within the territory of the reserve, there are 317 species of lichens, 121 species of mosses, and 810 species of vascular plants, of which 105 species are rare and require special protection, including 11 listed in the Red Book of the Russian Federation

and 27 in the Republic of Bashkortostan [13]. We suppose that our results can help reserve managers identify areas with high ecological value and prioritize conservation efforts. This includes safeguarding critical habitats, such as old-growth forests, which are essential for maintaining biodiversity.

Additionally, understanding GSV levels is crucial for participating in carbon offset programs and sustainable economic activities within and around the reserve. For instance, knowledge about forests can help estimate the carbon storage capacity of the reserve, contributing to global climate change mitigation efforts. It also enables the development of non-destructive activities such as eco-tourism, wild forest product harvesting (e.g., mushrooms, berries), and other nature-based enterprises that can support local communities.

## 5. Conclusions

Research of forest ecosystems is the most important task for environmental planning and response programs to climate change, especially over a long period. This paper presents the evaluation and digital mapping of forest growing stock volume (GSV) in Bashkiriya State Nature Reserve (Republic of Bashkortostan, Russia). Field observations were conducted in the process of forest surveys in 1979 and 2015. According to these materials, we identified the predominant tree species and calculated the GSV values. Then, we produced digital maps of GSV for each study year and calculated the GSV annual gain. The results showed that birch and pine were the dominant species, with an annual value of GSV of 1.4–2.1 $m^3$/ha per year in the studied area. Through a comparative analysis of GSV and annual growth, we have shown that the reserve has experienced a significant increase in GSV over the past 36 years. Produced digital maps can serve as a valuable tool for forest managers and policymakers in identifying areas of the reserve where forest management practices can be improved and where reforestation efforts may be needed. The findings of this study have important implications for further climate change mitigation as forests are a key carbon sink, absorbing and storing large amounts of carbon dioxide from the atmosphere.

**Author Contributions:** Conceptualization, L.B. and A.V.; methodology, A.V., A.S., I.G. and E.B.; software, A.V., A.S. and E.B.; validation, I.T. and R.S.; formal analysis, I.T. and R.S.; investigation, A.V. and L.B.; resources, L.B. and R.S.; data curation, A.V., L.B., I.G. and I.T.; writing—original draft preparation, L.B., A.S. and E.B.; writing—review and editing, A.V.; visualization, A.S. and E.B.; supervision, A.V., L.B. and I.G.; project administration, L.B., I.T. and R.S.; funding acquisition, L.B. All authors have read and agreed to the published version of the manuscript.

**Funding:** This study was funded by the Ministry of Science and Higher Education of the Russian Federation "PRIORITY 2030" (National Project "Science and University").

**Data Availability Statement:** The data presented in this study are available on request from the corresponding author.

**Conflicts of Interest:** The authors declare no conflict of interest.

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
