# Peer review of "Assessing and Mapping Changes in Forest Growing Stock Volume over Time in Bashkiriya Nature Reserve, Russia"

_forests, doi:10.3390/f14112251_

Round 1
Reviewer 1 Report
Comments and Suggestions for Authors
This research conducted a comparative analysis of forest growing stock volume (GSV) for 1979 and 2015, and determination of annual 15 growth in Bashkiriya Nature Reserve (Russia). Also, the paper evaluated the spatial distribution and produced digital maps of the species and their GSV for each year. This work has certain value and significance. But there are many problems in this paper that need to be further improved.
1. The abstract mentions "The increase in GSV in Bashkiriya Nature Reserve over the past 36 years represents a significant increase in carbon sequestration potential, which could help mitigate the effects of climate change. ", but the results in the analysis does not explain how forest climate change affect climate change. The authors either add relevant explanations or delete irrelevant expressions in the abstract.
2. Due to the lack of literature review of existing relevant studies, this paper should summarize the methods and results of other people's estimation of forest growing stock volume, and put forward the innovation points of this paper.
3. What are the advantages of conventional field methods over remote sensing data and machine learning methods? The paper should explain clearly the reasons for adopting conventional field methods.
4. The last part of the introduction puts forward the objectives of the paper, but the content written is about the research content, which does not reflect the marginal contribution of the study. The authors should outline the research purpose and the innovation of the paper.
5. There are some small details in the paper that need to be revised, such as why are there 3 "2.1"? Please check whether the serial number, formula and calculation results are accurate.
6. The application prospect of the results of GSV in this paper is not discussed. For example, the results of GSV can be used to monitor the effectiveness of forest ecological protection, the basis for ecological compensation, and so on. The content of the discussion part is too shallow and the depth is not enough. The authors should increase the comparison with other relevant research results and pay attention to exploring the innovative discoveries.
Comments on the Quality of English Language
The overall quality of the English language is good, but some details need to be further corrected.
Author Response
This research conducted a comparative analysis of forest growing stock volume (GSV) for 1979 and 2015, and determination of annual 15 growth in Bashkiriya Nature Reserve (Russia). Also, the paper evaluated the spatial distribution and produced digital maps of the species and their GSV for each year. This work has certain value and significance. But there are many problems in this paper that need to be further improved.
Dear reviewer 1. We grateful to you for reading our paper and your comments. Let us answer them.
All changes in the text were highlighted by yellow color.
- The abstract mentions "The increase in GSV in Bashkiriya Nature Reserve over the past 36 years represents a significant increase in carbon sequestration potential, which could help mitigate the effects of climate change. ", but the results in the analysis does not explain how forest climate change affect climate change. The authors either add relevant explanations or delete irrelevant expressions in the abstract.
Answer: Thanks for your valuable note. We agree. We eliminated this statement from abstract and conclusion part.
- Due to the lack of literature review of existing relevant studies, this paper should summarize the methods and results of other people's estimation of forest growing stock volume, and put forward the innovation points of this paper.
Answer: Thanks for the suggestions. We disclosed this part in the third paragraph of the introduction as “Currently, various methods are used to determine of GSV [3], and they can be mainly categorized as: 1) Sample tree measurements (including tree species, diameter at breast height, upper diameter, tree height and other); 2) volume models (estimate the sample tree volumes based on the previously described field measurements); and remote sensing technologies. Despite the fact that remote sensing data and machine learning methods are successfully used in GSV spatial assessment today [8–10], the correct implementation of these methods is severely limited without ground-based data. Thus, conventional field methods are necessary to establish relationships with explanatory variables (e.g., remote sensing data). This is especially important for previously unexplored areas, since there is no data for training statistical and mathematical models. In general, field measurements are highly detailed and accurate, making them particularly useful for training and validation of machine learning models. Nevertheless, should be noted that conventional methods have limitations, including resource and time requirements”. Please see lines 52-64.
In general, we emphasis, that although remote sensing data are popular today for GSV assessment, their application should be trained and validate using field measurements. Nevertheless, conventional methods have limitations, including resource and time requirements.
- What are the advantages of conventional field methods over remote sensing data and machine learning methods? The paper should explain clearly the reasons for adopting conventional field methods.
Answer: Thanks for your suggestion. We improve the introduction section in terms of your suggestion. We include the following sentences: “Despite the fact that remote sensing data and machine learning methods are successfully used in GSV spatial assessment today [8–10], the correct implementation of these methods is severely limited without ground-based data. Thus, conventional field methods are necessary to establish relationships with explanatory variables (e.g., remote sensing data). This is especially important for previously unexplored areas, since there is no data for training statistical and mathematical models. In general, field measurements are highly detailed and accurate, making them particularly useful for training and validation of machine learning models. Nevertheless, conventional methods have limitations, including resource and time requirements.”
We also emphasized this point in “4.3. Further prospects for digital mapping of tree species and GSV” section as: “Furthermore, having a dataset spanning different years (1979 and 2015) and collected using the same methodology, it is possible to utilize it for modeling GSV in both space and time, enabling us to forecast future changes.”
Please see lines 52-64 and 315-318.
- The last part of the introduction puts forward the objectives of the paper, but the content written is about the research content, which does not reflect the marginal contribution of the study. The authors should outline the research purpose and the innovation of the paper.
Answer: Thank for the suggestion. We have improved this part by add the following sentences:
“Currently, global forests are experiencing added stress due to climate change, which has a significant impact on GSV levels. Gschwantner et al. [3] emphasized in a review article that, to inform the future development of a knowledge-based bioeconomy, there is a demand for the maintenance, extension, and harmonization of existing forest databases. This is essential to enable the analysis of forest ecosystem changes on a large spatial and long temporal scale. In this regard, retrospective monitoring is a crucial solution for tracing changes in forest characteristics in both space and time.”
Please see lines 65-71.
We suppose that we better pointed the purpose and the innovation of the study.
- There are some small details in the paper that need to be revised, such as why are there 3 "2.1"? Please check whether the serial number, formula and calculation results are accurate.
Answer: Thanks for your notes. We have checked the paper and fixed various types of errors.
- The application prospect of the results of GSV in this paper is not discussed. For example, the results of GSV can be used to monitor the effectiveness of forest ecological protection, the basis for ecological compensation, and so on. The content of the discussion part is too shallow and the depth is not enough. The authors should increase the comparison with other relevant research results and pay attention to exploring the innovative discoveries.
Answer: Thank for the suggestion. We included key aspects of further use of GSV data in the discussion part as separate part “4.4. The prospect of applying GSV results”.
We thank you for your constructive comments. All changes in the text were highlighted by yellow color.
Reviewer 2 Report
Comments and Suggestions for Authors
In the manuscript, Bashkiriya Nature Reserve in Russia was selected as the study area to analyze the spatiotemporal variation of GSV and its driving factors, which can provide scientific basis for the establishment of forest carbon sequestration management measures in the Reserve. Judging from this viewpoint, the research content of this manuscript has a certain practical significance
However,this manuscript only uses the more conventional statistical analysis method , which is lack of significant innovation. At the same time, the research results of this manuscript belong to the commonsense in the field of forest resource monitoring, and lack of significant novelty. In the discussion section, the author spend a lot of space to discuss the application of remote sensing and machine learning in GSV monitoring, but these advanced methods are not involved in the manuscript.
Comments on the Quality of English LanguageMinor editing of English language required
Author Response
In the manuscript, Bashkiriya Nature Reserve in Russia was selected as the study area to analyze the spatiotemporal variation of GSV and its driving factors, which can provide scientific basis for the establishment of forest carbon sequestration management measures in the Reserve. Judging from this viewpoint, the research content of this manuscript has a certain practical significance
However,this manuscript only uses the more conventional statistical analysis method , which is lack of significant innovation. At the same time, the research results of this manuscript belong to the commonsense in the field of forest resource monitoring, and lack of significant novelty. In the discussion section, the author spend a lot of space to discuss the application of remote sensing and machine learning in GSV monitoring, but these advanced methods are not involved in the manuscript.
Answers:
Dear reviewer 2. We grateful to you for reading our paper and your comments. Let us answer them.
- Indeed, according to the objectives of our study, we focused our main efforts on comparing the two forest data sets presented for 1979 and 2015. According to our knowledge, the GSV retrospective studies are fragmentary (see https://doi.org/10.1016/j.foreco.2021.119868), which makes such investigations valuable for many reasons that we have indicated in the article (e.g., assessment of climate change influence, etc.).
We have improved and emphasized the novelty and importance of our study by adding the following sentences:
“Currently, global forests are experiencing added stress due to climate change, which has a significant impact on GSV levels. Gschwantner et al. [3] emphasized in a review article that, to inform the future development of a knowledge-based bioeconomy, there is a demand for the maintenance, extension, and harmonization of existing forest databases. This is essential to enable the analysis of forest ecosystem changes on a large spatial and long temporal scale. In this regard, retrospective monitoring is a crucial solution for tracing changes in forest characteristics in both space and time.”
Please see lines 65-71.
- Thanks for your note. Although the use of remote sensing methods for GSV evaluation is increasingly popular today, we did not set this goal (remote sensing and machine learning implementation). We had desire to pointed out that without ground-based observations impossible to conduct spatial modelling of forest properties. According to suggestions of another reviewer, we improved this section with the following paragraph: “Currently, various methods are used to determine of GSV [3], and they can be mainly categorized as: 1) Sample tree measurements (including tree species, diameter at breast height, upper diameter, tree height and other); 2) volume models (estimate the sample tree volumes based on the previously described field measurements); and remote sensing technologies. Despite the fact that remote sensing data and machine learning methods are successfully used in GSV spatial assessment today [8–10], the correct implementation of these methods is severely limited without ground-based data. Thus, conventional field methods are necessary to establish relationships with explanatory variables (e.g., remote sensing data). This is especially important for previously unexplored areas, since there is no data for training statistical and mathematical models. In general, field measurements are highly detailed and accurate, making them particularly useful for training and validation of machine learning models. Nevertheless, conventional methods have limitations, including resource and time requirements.”
Please see lines 52-64.
- We also expand our paper with emphasis on significant the results obtained. For instance, we included section call “4.4. The prospect of applying GSV results” in the discussion part, where we write that our fundings can 1) to contribute to sustainable forest management practices within the reserve; 2) to help reserve managers identify areas with high ecological value and prioritize conservation efforts; and 3) to involve carbon offset programs and sustainable economic activities within and around the reserve. Please see lines 319-340.
According to comments of other reviewers, we also improved other sections of paper. For example, we provided tables 1 and 2 divided by main regions of the reserve (Southern Krakka and Uraltay ridges) with information about square, GSV, forest species and their changes for the period 1979–2015. Also, we included description of these tables. Please see lines 152-182.
We hope that we have managed to improve the work and emphasize the significance of the results obtained. We believe that spatial-temporal research is essential, especially in the context of our rapidly changing environment and climate.
We thank you for your constructive comments. All changes in the text were highlighted by yellow color.
Reviewer 3 Report
Comments and Suggestions for Authors
The manuscript is very well written and is very easy for the reader to follow your study. However, there is a huge lack of content. The manuscript focuses on the description of two years 1979 and 2015, which is not the same as 36-year period and the entire document is very descriptive. The authors mentioned in the abstract “a significant increase in carbon sequestration potential” but it is not indicated how it has been calculated and past and future management measures that could promote it are discussed. On the other hand, there is no mention of the type of forest management carried out in the Bashkiriya Nature Reserve or possible assumptions or measures to improve growing stock volume (GSV). In addition, the study area should be divided since there are great differences between areas dominated by different species and the age classes that have been considered should be defined. Therefore, beyond the descriptive histograms presented, I have not found any results or proposals that are sufficient for the publication of this study.
Author Response
The manuscript is very well written and is very easy for the reader to follow your study. However, there is a huge lack of content. The manuscript focuses on the description of two years 1979 and 2015, which is not the same as 36-year period and the entire document is very descriptive. The authors mentioned in the abstract “a significant increase in carbon sequestration potential” but it is not indicated how it has been calculated and past and future management measures that could promote it are discussed. On the other hand, there is no mention of the type of forest management carried out in the Bashkiriya Nature Reserve or possible assumptions or measures to improve growing stock volume (GSV). In addition, the study area should be divided since there are great differences between areas dominated by different species and the age classes that have been considered should be defined. Therefore, beyond the descriptive histograms presented, I have not found any results or proposals that are sufficient for the publication of this study.
Answers:
Dear reviewer 3. We grateful to you for reading our paper and your comments. Let us answer them.
- Regarding the “a significant increase in carbon sequestration potential” sentence:
Thanks for your valuable note. We agree. We eliminated this statement from abstract.
- Regarding the type of forest management.
We clarified and included the following information: The designation of the territory as a "state natural reserve" implies a complete cessation of economic use. Only data collection and observation of ecosystem changes are permitted. However, it should be noted that the reserve was closed in 1951 and resumed operation only in 1958. During this period, a forestry enterprise operated within its boundaries, with timber harvesting as one of its responsibilities. Additionally, cattle grazing took place in the mountainous steppes, parts of the territory were plowed for agriculture, and extensive areas were used for haymaking. Nevertheless, in 1958, the reserve status was reinstated, and today only scientific observations have been con-ducted within its territory. Please see lines 100-107.
- There is no possible assumptions or measures to improve GSV
Thanks for your note. In Section “4.3.Further prospects for digital mapping of tree species and GSV” we a discuses the possible ways to improve GSV assessment, including remote sensing data (e.g. LiDAR) or unmanned aerial vehicles (UAV). Moreover, according to other reviewers’ comments, we added the section “4.4. The prospect of applying GSV results”. Please see lines 319-340.
- The study area should be divided since there are great differences between areas
Thanks for the valuable suggestion. We provided tables 1 and 2 divided by main regions of the reserve (Southern Krakka and Uraltay ridges) with information about square, GSV, forest species and their changes for the period 1979–2015. Also, we included description of these tables. Please see lines 152-182.
We hope that we have managed to improve the work and emphasize the significance of the results obtained. We believe that spatial-temporal research is essential, especially in the context of our rapidly changing environment and climate.
We thank you for your constructive comments. All changes in the text were highlighted by yellow color.
Round 2
Reviewer 1 Report
Comments and Suggestions for Authors
According to the opinions, the author has made necessary modifications and improvements to the paper. But as far as research methods are concerned, there is still a small problem. The author mentioned in the paper that there are limitations in conventional field methods, and how to avoid such limitations is very important, which should also be an important innovation point in the paper. If the author does not address the limitations of this method in this paper, the discussion section should discuss how to address the limitations of traditional methods in the future.
Author Response
Dear reviewer 1. We are glad to hear that you have appreciated our improvements.
In our work, we emphasized that conventional methods, although they are the most reliable and important for future forest research, they remain time-consuming and expensive.
In general, in section 4.3.”Further prospects for digital mapping of tree species and GSV” we discussed how to address the limitations of conventional methods. Thus, we emphasis on apply remote sensing data and machine learning approaches. Moreover, we updated this part and included the following thoughts: “In the future, addressing the limitations of conventional methods for GSV assessment could involve embracing technological advancements and innovative approaches…”
“Also, one key solution is to apply volume models, which estimate sample tree volumes based on previously described field measurements. This approach can significantly reduce the time and resources required for extensive fieldwork in similar ecosystems…”
“These approaches, combined with ongoing research and technological advancements, offer promising avenues for overcoming the limitations of traditional methods in forest growing stock assessment…”
Thanks for reviewing our work and your valuable comments,
Best regard, team of authors
Reviewer 2 Report
Comments and Suggestions for Authors
In response to the comments raised by the reviewers, the author gave a serious reply, and in the revised manuscript, the author supplemented and improved some contents including the research objectives and discussion. However, two problems still exist in the revised manuscript as following: apart from conventional statistical analysis,the research method in the manuscript lacks significant innovation; the research results do not propose new findings that are beyond common sense in the forestry community. Based on this, the paper does not reach the level of publication in the international journal of Forests.
Comments on the Quality of English LanguageMinor editing of English language required
Author Response
Dear reviewer 2. We are glad to hear that you have appreciated our improvements.
We do not dispute that our work claims to be breakthrough discoveries in the forest science. Nevertheless, we believe that retrospective studies deserve attention to be presented to the scientific community. We examined how the mechanism of biomass growth changes in an entirely undisturbed area.
We did not specify the set of data used in the article: so, in 1979, 8395 and 8405 forest plots were surveyed, respectively. We consider this to be a significant set of data and changes in GSV values over this period can be considered as reliable. We added this information in abstract and methods part.
We hope that our study leveraged a substantial dataset, which provided valuable retrospective insights into the dynamics of forest ecosystems, allowing for the assessment of changes over a 36-year period. These aspects add depth and reliability to your research, making it a valuable contribution to the field of forest ecology and environmental monitoring. In addition, we draw your attention to the fact that the research was conducted in a territory that has not undergone transformations caused by human activity for 65 years. The results of our study may be useful for other researchers engaged in the study of the dynamics of forests in economic use. We also additionally analyzed the dynamics of not only forested territories, but also steppes, as well as former hayfields and areas of burnt forest (lines 185-193) and found out that the renewal of the forest occurs precisely in the territories transformed in the past, while natural steppe ecosystems also expand their participation in the structuring of the landscapes of the Bashkir State Reserve.
We can call this study as “case study”, and we believe that it also deserves to be presented to the scientific community.
Thanks for reviewing our work and your valuable comments,
Best regard, team of authors
Reviewer 3 Report
Comments and Suggestions for Authors
I positively value the changes made by the authors in the previous manuscript and believe that they have substantially improved the new version of the manuscript. For this reason, despite the fact of not having obtained results that allow you to draw new conclusions, the study in my opinion, is now suitable for publication.
Author Response
Dear reviewer 3! Thank you very much for evaluating the work we have done!
Best regard, team of authors